causalizeR: a text mining algorithm to identify causal relationships in scientific literature

Ancin-Murguzur Francisco J. francisco.j.murguzur@uit.no
Hausner Vera H.
The Arctic Sustainability Lab, UiT the Arctic University of Norway , Tromsø , Norway
Piccolo Stephen
Electronic publication date: 2021 Jul 20
Publication date: 2021
Volume: 9
Electronic Location ID: e11850
Received 2021 Feb 11; Accepted 2021 Jul 2
Copyright: ©2021 Ancin-Murguzur and Hausner
Copyright year: 2021
Copyright holder: Ancin-Murguzur and Hausner
License: This is an open access article distributed under the terms of the Creative Commons Attribution License, which permits unrestricted use, distribution, reproduction and adaptation in any medium and for any purpose provided that it is properly attributed. For attribution, the original author(s), title, publication source (PeerJ) and either DOI or URL of the article must be cited.
License URL: https://creativecommons.org/licenses/by/4.0/

Keywords: Big data, Evidence synthesis, Scenarios, Natural language processing, Literature review

Funding: Fram Center Flagship Effects of Climate Change on Ecosystems, Landscape Local Communities and Indigenous People 369903 Project EcoShift 296987 Future ArcTic Ecosystems (FATE) UiT The Arctic University of Norway This article was supported by the Fram Center Flagship Effects of Climate Change on Ecosystems, Landscape Local Communities and Indigenous People grant nr. 369903, Project EcoShift: Scenarios for linking biodiversity, ecosystem services and adaptive actions and the Norwegian research council grant nr. 296987, project Future ArcTic Ecosystems (FATE): drivers of diversity and future scenarios from ethnoecology, contemporary ecology and ancient DNA The publication charges for this article have been funded by a grant from the publication fund of UiT The Arctic University of Norway. The funders had no role in study design, data collection and analysis, decision to publish, or preparation of the manuscript.

==============================
Complex interactions among multiple abiotic and biotic drivers result in rapid changes in ecosystems worldwide. Predicting how specific interactions can cause ripple effects potentially resulting in abrupt shifts in ecosystems is of high relevance to policymakers, but difficult to quantify using data from singular cases. We present causalizeR (https://github.com/fjmurguzur/causalizeR), a text-processing algorithm that extracts causal relations from literature based on simple grammatical rules that can be used to synthesize evidence in unstructured texts in a structured manner. The algorithm extracts causal links using the relative position of nouns relative to the keyword of choice to extract the cause and effects of interest. The resulting database can be combined with network analysis tools to estimate the direct and indirect effects of multiple drivers at the network level, which is useful for synthesizing available knowledge and for hypothesis creation and testing. We illustrate the use of the algorithm by detecting causal relationships in scientific literature relating to the tundra ecosystem.

Introduction

One of the major challenges for ecosystem scientists is to unravel the complex interactions among drivers that can explain how ecosystems are responding to global warming and human impacts (LaDeau et al., 2017; Peters & Okin, 2017). Synthesizing evidence from literature is a necessary step to identify potential drivers and interactions that could influence changes in ecosystems as a result of climate change (Hassani, Huang & Silva, 2019). Due to the large amount of work necessary to extract the relevant information from each published article, most evidence syntheses narrow down the focus to a limited set of interactions in ecosystems that will change as a result of the accelerating global changes. Natural language processing tools offer the opportunity to classify literature into topics based on the words that are contained in the text (Blei, Ng & Jordan, 2003; Syed & Weber, 2018). Topic modeling has been used for ecology and conservation (Han & Ostfeld, 2019; McCallen et al., 2019), but the algorithm used do not fully extract the detailed information contained in the text. Other tools are therefore needed to synthesize scientific evidence across multiple cases to extract information about interacting drivers that can alter ecosystem patterns and processes (Eskelinen et al., 2020; Lawrence et al., 2007). A research synthesis built on automated content analysis could benefit ecosystem science by detecting links between drivers and build scenarios that can capture the full range of plausible ecosystem interactions.

Text mining tools have been applied in life sciences such as medicine (Tsuruoka et al., 2011) or molecular biology (Miwa et al., 2009), but its application has primarily been used in the biomedical disciplines (Ciaramita et al., 2005; Quan, Wang & Ren, 2014). These algorithms require a rigid structure of texts and a targeted search of relations between concepts. Ciaramita et al. (2005) used a set of relational keywords to establish relationships such as virus encode protein, and parsing it to a pre-established structure in the text. This approach does not allow the flexibility needed to extract relationships from unstructured texts, where the interactions between drivers and ecosystems are unknown or challenging to identify (Mirza & Tonelli, 2016). Scientific and gray literature are clear examples of unstructured texts that contain large amounts of information that are not easily accessible for text mining algorithms. Therefore, an algorithm that can interpret the relational contents of these texts is necessary in order to expand the current knowledge.

In natural sciences, the ability to extract causal links between drivers in the ecosystem from available literature combined with network analyses can help to improve our understanding of the system and the direct and indirect effects of drivers. The strength of such a method is the creation of a systematic interaction map over drivers and ecosystem effects, detection of previously unknown causal links in ecosystems, and facilitate scientific learning by drawing on the available literature for conceptualizing ecosystem models. This interaction map does not provide evidence of causes and effects, but provides an indication of which drivers are most often commonly referred to in the text as influential on an ecosystem component. The mechanisms behind the effects of drivers (i.e., why is the change happening) are not captured by text mining approaches, as they require a more systematic review of the evidence that is given by literature reviews.

In this study we created a natural language processing algorithm that extracts causal links between different drivers of change from texts. We applied the algorithm to a database with 9274 abstracts containing the keyword “tundra” from the Scopus database (Ancin-Murguzur & Hausner, 2020) and searched for strong interaction indicator words (increase, promote, decline and decrease). Based on the output from the algorithm, we developed a causal relational map (similar to a food web) using social network analysis tools that identifies the main drivers and interactions in tundra ecosystem and discuss the applicability of the algorithm combined with social network analysis tools.

Proposed Method

Our proposed method is summarized in Fig. 1: the algorithm uses the text annotation abilities of the package udpipe in R (Wijffels, 2019), giving a grammatical structure to each sentence. More specifically, we make use of the class to which each word belongs to (e.g., noun, verb, adverb, …) to extract the information related to drivers. In nature, a parameter identified as a driver can also be influenced by other parameters, which is reflected in the complexity of natural interaction networks: for simplicity, all parameters (words) are therefore called “component” in this text.

Figure 1 Workflow for text analyses.

The diagram shows the suggested workflow. The literature database size will determine the amount of links that the causalizeR algorithm will detect and the extent of the resulting network for further analyses.

After the text annotation, the algorithm applies two basic grammatical rules to extract the causal relationship between two components:

• Rule 1 (Active voice): component A affects component B (e.g., grazing increase soil nutrient concentration)

• Rule 2 (Passive voice): component B is affected by component A (e.g., soil nutrient concentrations are increased by grazing)

The algorithm detects the effect of interest that has to be explicitly stated before processing the text (e.g., increase, decrease, promote, decline), and extracts the nouns that are immediately before and after it, assuming the general structure of “NOUN- verb-NOUN”. Since it is common that the components are composed by two nouns (e.g., soil carbon), the algorithm also finds if there is another noun adjacent to the detected one, allowing for the structure “NOUN-NOUN-verb-NOUN-NOUN” if two nouns are adjacent to each other. However, there are several instances where the text annotation toolkit wrongly assigns nouns as adjectives when there are two consecutive nouns, as well as some individual words such as herbivory, although herbivore is considered a noun, thus the algorithm allows to detect “(NOUN/ADJ)-NOUN-verb-NOUN-(NOUN/ADJ)”. The algorithm ignores other word types such as adverbs, retrieving a consistent database of only components related to the target verb.

Each component is assigned to the “driver” or “affected parameter” category at each sentence, depending on the rules mentioned above. This process is performed on a sentence-to-sentence basis, and only accounts for the first appearance of the effect of interest in a sentence. The effect of the driver can be assigned to a value depending on its direction (increase and promote can be assigned a +1 value, while decrease and decline can be assigned a -1 value) or any other numerical value to the link (e.g., a weight) if the effect of interest can be given a more precise weight (e.g., slightly increase can be assigned a weight of +0.5 to reflect a weaker effect than increase).

Data post-processing

A manual word cleanup is needed after the algorithm has processed the texts to remove clearly nonsensical relationships that arise due to word placement, e.g., “our results show an increase in productivity” will return the causal relationship of results-increase-productivity, which does not provide us with useful information due to lack of context. Word harmonization such as synonyms or closely related words can create a more simplified causal relationship map that allows for an easier interpretation of the results; however, word harmonization and concept pooling (i.e., merging words with similar concept) can be adjusted to the focus of a study to highlight the relevant components to answer different specific hypotheses (e.g., pooling all herbivores into grazing, or group herbivores into ungulates and rodents). Repeated relationships between the same concepts are pooled into a single relationship by averaging the effects to account for opposing results.

Some words have no meaning without context and should be deleted before using the network or even creating the network e.g., symbols such as %, which often appears when talking about effect sizes. These cases need to be individually considered and decided if that word or symbol warrants elimination prior to running the algorithm to extract more meaningful information.

Network creation and analyses

After the database harmonization, we created a network based on social network visualization tools from the qgraph package (Epskamp et al., 2012). These tools create a network that shows the links between the different parameters in the system, assigning the strength and direction of the link (e.g., a strong positive effect of a component on another). These networks can be used to confirm and expand or create hypotheses by exploring pathways between links. This network can be used to understand how different components interact and to create data-driven hypotheses in an incremental manner.

In addition to the main network, which is very complex to assess visually, we created a data-driven hypothesis starting from reindeer as the main research subject: for that purpose, all mentions to reindeer (e.g., reindeer population or reindeer grazing) were converted to reindeer: we did not post-process the dataset any further to illustrate the raw result obtained from the algorithm. We then selected one of the linked components (soil N) to show how can a hypothesis be formed based on the interaction matrix, visualized with SNA tools.

Results

The algorithm created a causal relationship database with 7,540 links between components, resulting in a highly complex and intertwined network. The network that shows reindeer as the component of interest show clear effects of reindeer in the tundra, such as increase in soil N and decrease in soil P, which is explained by the fecal deposition of N or the intake of P for antlers (Sitters et al., 2019) (Fig. 2).

Figure 2 Network representation of the components directly related to reindeer in the tundra ecosystem.

Network representation of components affecting and affected by reindeer, as extracted from the main network. Dark solid arrows indicate a positive effect and grey dashed arrows indicate a negative effect.

Expanding the reindeer interaction network one step further by adding the linkages to soil n, which is a limiting factor for primary productivity (Shaver & Chapin, 1980), creates a more complex network with explicit links between ecosystem components. Here, we can see that microbial activity has a positive impact on soil N, and that soil N favors plant productivity and root density (Fig. 3), but other components can be more difficult to interpret such as fold (Fig. 3) or question, which should be carefully considered for deletion or manual correction.

Discussion

The combination of the algorithm and the network analysis tools opens new avenues in research. Our algorithm extracts causal links reported in published literature by means of simple grammatical rules that offer high flexibility to process unstructured texts and identifies the interactions between the components of ecosystem change. Opposite to the time it requires for humans to do a literature review to develop a similar database, the algorithm perform this process automatically resulting in a rapid database development. The visualization of ecosystem interactions helps understand how components can affect ecosystem patterns and processes which can be used to construct hypothetical scenarios of ecosystem changes in which the direct and indirect effects of variations at different levels of the network can be analyzed.

The keyword selection is the most important aspect of this process, as the algorithm does not directly understand the text, it rather works on the location with respect to a keyword: in this case, we used strong causational indicators (increase, promote, decline and decrease), while other modulators and word combinations could reveal more specific interactions between components (e.g., predate for predator–prey interactions). The algorithm identified 7540 unique links between components in the tundra literature example, which allowed us to create a database of the major causal relationships in the tundra ecosystem. The combination of the algorithm and network visualization tools result in a large-scale food web and its interactions with the environment. The alternative of a manual approach would require long periods of reading texts by scientists but would also reduce noise created by nonsensical word relationships. There is a trade-off between time and precision, as manually creating these links would result in a more coherent dictionary of components, although the strongest links will be mentioned more than once in literature, which makes it highly unlikely that the algorithm will fail to recognize those links due to writing styles. Furthermore, this algorithm could be used for analyzing full texts and grey literature, thus including a much larger body of source information.

Figure 3 Network representation of the components related to reindeer and soil N in the tundra ecosystem.

Network representation of components affecting and affected by reindeer and soil N, as extracted from the main network. This network shows how an incremental approach can help understand the system in a stepwise fashion. Dark solid arrows indicate a positive effect and grey dashed arrows indicate a negative effect.

The ecosystem linkages derived from the algorithm could be constructed by focusing on a single component of interest and construct an interaction network automatically from literature (Fig. 3), which in turn could be used to develop new hypotheses and understand potential indirect interactions between components. Furthermore, the network based on all the interactions (the main network) can be used to investigate the pathways between components and explore indirect effects of changes in components. Furthermore, the network can also be incorporated in a fuzzy cognitive mapping (Dexter et al., 2012) or Bayesian network modeling pipeline for scenario creation under climate change, extreme events or management actions to estimate an expected outcome.

In general, the causal links present in abstracts are straightforward to interpret, but other keywords are more ambiguous, and can be prone to be misinterpreted. For example, the letter “c”, could refer to carbon or to centigrade (as in oC): this issue is somewhat solved by looking for the noun (or adjective) immediately before the letter “c”, which can give context (e.g., “degree C” or “tons C”). In any case, these inconsistencies will always be present in automated approaches, as algorithms nowadays are unable to elicit the meaning of a word or acronym from the context. The algorithm registers the text where the interaction was detected, so that it is possible to manually assess the text if that is deemed necessary. While a manual adjustment is necessary for a set of links and concepts, using automated methods to analyze large databases reduces biases due to subjective interpretation of texts and exhaustion of the operators doing repetitive tasks (Gonzalez et al., 2011; Healy et al., 2004). Using a two-noun registration as opposed to using only the first noun immediately before and after the target word gives more complexity to the resulting network, but allows to adjust the level of details by manually clustering concepts together depending on the use given to the causal database: general, ecosystem-wide projects will benefit from simplified concepts (e.g., grouping together animals by trophic groups for a simplified food web description) whereas specific interactions will aim for more detailed interaction network (e.g., the relation of a given rodent species with a predator), or a combination of simplified components for a part of the ecosystem and detailed for another (e.g., simplified abiotic and detailed biotic drivers).

Conclusions

We present an automated algorithm to extract causal relationships from large corpora of texts and apply it to the tundra literature as a case study. The use of the algorithm is not limited to abstracts or scientific texts. This tool allows researchers and students to create directed networks that can further be used in management scenarios to take better informed decisions, and to identify the indirect effects of changes in individual components of the ecosystem.

Supplemental Information

Supplemental Information 1 Dataset and script to install the R-package and replicate the results in the article

The data folder includes the bibliographic data, the script used to analyze the data and a snapshot of the R-package used for this publication. The R-package may be updated in the github repository with possible bug fixes and new features.

Click here for additional data file.

Additional Information and Declarations

Competing Interests

Author Contributions

Data Availability

The authors declare there are no competing interests.

Francisco J. Ancin-Murguzur conceived and designed the experiments, performed the experiments, analyzed the data, prepared figures and/or tables, authored or reviewed drafts of the paper, and approved the final draft.

Vera H. Hausner conceived and designed the experiments, prepared figures and/or tables, authored or reviewed drafts of the paper, and approved the final draft.

The following information was supplied regarding data availability:

A copy of the R package, together with the data and a commented script used to generate the results and figures, are available in the Supplemental Files.

The algorithm is available as an R package named “causalizeR” at https://github.com/fjmurguzur/causalizeR; Zenodo: fjmurguzur. (2021, May 27). fjmurguzur/causalizeR: causalizeR (Version v1.0-beta). Zenodo. http://doi.org/10.5281/zenodo.4817639; and at UiT open research data: Ancin-Murguzur, Francisco Javier; Hausner, Vera Helene, 2021, “Replication Data for: causalizeR: A text mining algorithm to identify causal relationships in scientific literature”, https://doi.org/10.18710/PTQ8X7, DataverseNO, V1.

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
