# Peer review of "causalizeR: a text mining algorithm to identify causal relationships in scientific literature"

_PeerJ, doi:10.7717/peerj.11850_

## Round 0.1 · original submission · Major Revisions

The reviewers have provided helpful comments. Please address their comments carefully and send us a revision.

Reviewer 1 ·

Basic reporting

I took a few days to read this manuscript as it deals with an area that I've been working in for the past couple of years, and I must say that the article is well written, with clear and concise language, and I thoroughly enjoyed reading it. There are a few minor comments about the basic reporting that I believe the authors can work on to improve their article.

Minor corrections to the grammar in the article can help convey the design and findings of the work convincingly, such as corrections on line 43 (a data-driven, not an),.

In the introduction, the authors claim that within the broad area of biological research, NLP tools have seen applications primarily in biomedical disciplines. However, multiple areas within the broad umbrella of biological research have witnessed the application of NLP to scientific literature, such as ecology (Han & Ostfeld 2019), and conservation (see: McCallen, Knott et al. 2019; Nunez-Mir, Iannone et al. 2016; Knott, Rue et al. 2016). It would be prudent of the authors to cite these works as well.

Experimental design

The authors have clearly described the framework of their experimental setup in the final few lines of the introduction itself. Immediately, I know what to expect from the following sections and there is no ambiguity here and the authors must be congratulated on their writing!

In the "Proposed Design" section, authors have also taken into consideration several areas where the language of the texts that they'll be analyzing might adversely affect the outcomes of their analysis and have accordingly accounted for such outliers in their dataset.

Overall, the experimental design of the study makes sense. The authors have broken down this complex problem into a set of logical rules and analyzed this simplified state of the texts, rather than the nuanced language used in scientific texts.

Validity of the findings

The authors might want to briefly explain some of the jargon associated with network analysis, before diving into their discussion of results. For example, I had to look up the meaning of casual relationships, linkages and unique links in the context of network analysis before diving into the results.

The presentation of the figures could be improved. The authors might consider using Graphviz to generate these figures, since they're essentially bi-directional graphs. The authors might also consider improving the resolution of their figures. The labels in Figure 1 at the top left corner below the keyword "height" are not legible, and might be relevant to readers of the article upon publication. The overlap between the keywords and edges of the graph in Figure 2 make it difficult to appreciate the intuitiveness of the results presented.

Additional comments

As mentioned earlier, I thoroughly enjoyed reading your article! I really look forward to reading more about this work, especially on this very exciting bit put forth in the Discussion about scenario creation. One could analyze vast troves of scientific literature say, pertaining to climate change and develop food webs such as the one in Figure 2, and analyze and dissect the factors leading to climate change from literature. Such tools will aid researchers in identifying potential "breadcrumbs" in the literature that they might not have initially posited.

This is just one of the many possible applications of your tool, in addition to management scenarios that you have mentioned under Conclusions. Such tools can be invaluable to both students embarking on their research, as well as to advanced researchers identifying causal relationships in lesser studied ecosystems etc. I hope you broaden the scope of the applications of your tool in the Conclusions.

The suggestions that I've put forth are minor ones and I hope that you consider them in good sport. All the best with this work and I hope to see more of this research in the near future.

Reviewer 2 ·

Basic reporting

The reporting is clear and unambiguous. Please see comments below.

Experimental design

The methods needs a bit of work and I have outlined where the authors can focus on.

Validity of the findings

See comments below.

Additional comments

Overarching comments:

While automated content analysis tools can potentially offer a causal relationship or even establish a causal relationship simply based on the overall co-occurrence of certain keywords/nouns (in this case), ecologists are often interested in asking why there might be an underlying relationship or the basis for a particular complex ecosystem interaction. Identifying a complex interaction might not necessarily need a text mining approach. Could the authors make a stronger case why a text mining approach rather than a standard literature review would help unravel any complex interaction?

I would recommend that the authors add an additional paragraph to establish the novelty of their package – which is combining network analysis and text mining. Perhaps start by talking about what network analysis is, in the first place and establish the strength of combining the two.

Sievert and Shirley (2014) wrote a package in python called pyLDAvis which essentially creates a concept map that showcases intertopic distance between topics/clusters of words that are often combined. How is your package unique/different from the above?

Can the authors provide a figure that is a flowchart of their package so that the readers understand what the authors have tried to showcase here?

As I read the manuscript, I wonder if it maybe easier for the reader to understand the workflow first, and then the application – where the authors should reiterate what their hypothesis is before they showcase how the hypothesis is being tested using their framework.

Section-specific comments:

Please provide a link to the package/associated code in the abstract (like a Methods in Ecology and Evolution article for the reader to get a sense of the applied aspect of the package).

Line 74: Where were the abstracts sourced from?

Lines 169-171: Can probably find a better place in the introduction.

---

## Round 0.2 · Minor Revisions

Thank you for addressing the reviewers' comments. In my view, the only remaining task that you must address is to make the online repositories publicly available. This is a requirement for publication at PeerJ, and I have seen too many times when authors promise to do this but never do it after the paper has been accepted.

---

## Round 0.3 · accepted · Accept

Thank you for making your code and data available. Please do add a link to the Zenodo repository to the manuscript. Rename the "R-Package availability" section to "Code and data availability" and add the Zenodo link there. Please also make sure the URL to the data is a link.